# Mechanical Properties and Toxicity Risks of Lead-Zinc Sulfide Tailing-Based Construction Materials

**DOI:** 10.3390/ma14112940

**Published:** 2021-05-29

**Authors:** Yang Zhou, Xinlian Duan, Tao Chen, Bo Yan, Lili Li

**Affiliations:** 1State Key Laboratory of Organic Geochemistry, Guangzhou Institute of Geochemistry, Chinese Academy of Sciences, Guangzhou 510640, China; zhouyang2@gig.ac.cn; 2University of Chinese Academy of Sciences, Beijing 510049, China; 3School of Environment, South China Normal University, University Town, Guangzhou 510631, China; 2019023440@m.scnu.edu.cn (X.D.); bo.yan@m.scnu.edu.cn (B.Y.); 4Guangdong Provincial Key Laboratory of Chemical Pollution and Environmental Safety & MOE Key Laboratory of Theoretical Chemistry of Environment, SCNU Environmental Research Institute, South China Normal University, Guangzhou 510631, China; 5College of Environmental Science and Engineering, Zhongkai University of Agriculture and Engineering, Guangzhou 510408, China; lilili@gig.ac.cn

**Keywords:** leaching residue of the lead-zinc sulfide tailing, construction materials, strength and stability, process mineralogy

## Abstract

The leaching residue of the lead–zinc sulfide tailing (LRT) is the only residue generated from the tailing leaching recovery process; it is a typical hazardous material for its high heavy-metal contents and high acidity. Due to the large output of LRT, and because its main components are Ca, Si, and Al, the preparation of building construction materials with LRT was studied. The results showed that when the LRT addition is less than 47%, with the ordinary Portland cement (OPC) and fly ash (FA) added and the curing conditions appropriate, the strength values of the tested specimens meet the M15 Class of the autoclaved lime sand brick standard (GB/T 16753-1997). The carbonization coefficient and drying shrinkage of the specimen were 0.79 and smaller than 0.42, respectively. As the SEM, TG, and XRD analysis have shown, the LRT can chemically react with additives to form stable minerals. The heavy metal contents that were leached out well met the limits in GB5085.3-2007. Based on the high addition of the LRT, the good strength and lower heavy metals were leached out of the prepared test specimen, and the tailing could be reused completely with the leaching recovery and the LRT reuse process. LRT can be used to replace OPC, allowing more sustainable concrete production and improved ecological properties of LRT.

## 1. Introduction

Millions of tons of Pb–Zn tailings have been stored in tailing impoundments without any protection. When exposed to natural weathering and chemical percolation, the heavy metals, including Cd, Co, Cr, Cu, Mn, Ni, Pb, and Zn, become more soluble and mobile. Once the heavy metals are leached, they become a major source of contamination. On the other hand, due to its huge amount, the utilization of tailings has great potential for reuse of the resource. Metals could be recycled from the tailings with separation and intensive leaching process. While, after metals recovery from Lead–Zinc Sulfide Tailings, a large amount of leaching residue of Lead–Zinc Sulfide Tailings (LRT) will be produced [1], and little attention has been paid to its disposal so far. The LRT output was varied based upon the different leaching recovery procedures [2,3], and its storage was stint by its acidity and the heavy metals. LRT was one classical type of solid waste, and its main components are elements such as Si, Ca, Al, and Fe [4,5,6]. These elements are just the main components of construction materials [5,7]. The recycling of the residue could not only realize zero emissions from the tailings’ recovery but would also offer a huge material for the building industry, which is a more effective resource recovery alternative [8,9,10,11,12]. Furthermore, its environmental risks would be effectively addressed by the construction material curing process [13,14,15,16]. Once the strength of the specimen with LRT added reaches the limit of the MU15 class construction standard (the compressive strength is 15MPa after 28 days curing process), it can be reused as foundations and other buildings [17]. In the mining area, there are varied and huge demands for this material, such as the reclamation of the tailings impoundments, the hardening of the mining area, and the filling for mining wells. In conclusion, the building materials prepared from the LRT can be fully used, which also solves the problem of stacking. However, there are some drawbacks during the recycling process. At first, secondary pollution, such as acidic gas (SO_2_, HCl, and so on), might be generated during the incineration process [18]. Secondly, the strength of the materials produced is not enough for extensive usage [19]. Other than that, the formed chemical composition of the minerals is unstable and binds water contained, the building materials have poor thermal stability, and they crack and age easily [20]. The objectives of the study were to produce better qualified and environmentally friendly construction materials with the residues generated from the leaching process.

## 2. Materials and Methods

### 2.1. Experimental Approach

The source of the tailing used in this study was the Fan-Kou lead–zinc sulfide mine, and the leaching LRT came from the end of the flotation separation process and the leaching process [5]. The LRTs were washed with distilled water and dried at 105 °C for reuse. Pulverized fly ash (PFA) was produced at local power plants, and ordinary Portland cement (OPC) was purchased from local Conch brand cement.

### 2.2. Building Construction Materials Preparation

The LRTs were added at a different rate, from 0 to 58% (W/W); the variation is detailed in Table 1. Mixtures with different proportions of LRT, pulverized fly ash (PFA), and ordinary Portland cement (OPC) were mixed in a stainless steels blender until homogeneous. The specimens were prepared and molded in a cube with a length and width of 50 mm and a height of 50 mm. Then, the specimens were demolded and cured in a moist cabinet at 90% humidity and at 45 ± 2 °C. The compressive strength was measured by a press testing machine (YES-100, YES TECH, Changsha, China) at 3, 7, and 28 days. Flexural strength was measured by the hydraulic pressure testing machine (DYE-300KN, Zhejiang, China). All dates of the strength were the arithmetic mean of six experimental data, and the values of the tested specimens were compared with the autoclaved lime sand brick strength standard of China(GB/T 11945-2019) [21].

### 2.3. Characterization of Building Construction Materials

Morphologies of the cured specimens were examined through scanning electron microscopy (SEM; JSM-5610LV, JEOL Ltd., Tokyo, Japan). The tested samples were used in X-ray diffraction (XRD, Rigaku, Tokyo, Japan) analysis to determine the mineral compositions. The thermal stability of the materials was measured in a thermal analyzer (STA 449 F3, NETZSCH, Selb, Germany). A leaching test was performed according with sulfuric acid and nitric acid method. The specimens were crushed so they could pass through a 9.5 mm nylon sieve, and then they were leached by a mixed acid solution (HNO_3_ and H_2_SO_4_, Analytically pure, Nanjing Chemical, Nanjing, China) with a pH of 3.20 ± 0.05 for about 18 ± 2 h in a turnover type shaker (YKZ-08, Yonglekang, China). An Inductively Coupled Plasma Mass spectrometry (ICP-MS, Agilent7700, Agilent, Washington, DC, USA) was used to determine the amounts of Cu, Pb, Zn, Cr, Cd, and Ni that had leached. Major elements such as Ca, Al, Mg, K, Na, Si, P, and Si were tested with X-ray fluorescence (XRF, S2polar, Bruker, Karlsruhe, Germany). In addition, to the pH measurement, the test procedure adopted in the previous literature [22] was considered, in which the tested samples of the cubes after 28 days of compressive strength were crushed into powder, and the sample was mixed with distilled water (1:10) in a beaker. The mixture was stirred well, and then the filtered solution was tested for pH.

## 3. Results and Discussion

### 3.1. Characteristic of the Tailing

As shown in Table 2, with the 73.86% of SiO_2_, 3.16% of Al_2_O_3_, and 2.72% of CaO, the main component of the LRT was similar to the construction materials. The contents of the targeted major metals such as Cu, Zn, Fe, and Mn were 2923 g/t, 1638 g/t, 108,671 g/t, and 591 g/t, respectively. The content of Na and K were lower than 0.2%, and S was mainly in the presence of sulfate; thus, the cracking risks caused by the dissolved salt and acidity will be reduced [18]. When the firing process was used, the available sulfur was easily turned to the notorious SO_2_ [18]; therefore, this study planned to use curing methods to prepare building specimens.

### 3.2. Specimen Preparation and Its Mechanical Strength Properties Analysis

The LRT was blended with OPC and FA in appropriate proportions to prepare concrete (Figure 1). The preliminary appearance structure and molding observation had shown that the preparation of cemented building materials with LRT was feasible. FA was selected for its ability to improve flow ability and for the paste’s mechanical properties [23]. The proportion of the FA and LRT with a diameter smaller than 150 μm was 99.03% and 85.96%, respectively, the median diameters (D50) were 11.41 and 97.16 μm, respectively. The values for the compressive and flexural strengths of the specimens prepared with different ratios of LRT and OPC are shown in Table 3. A 42.5 ordinary OPC was selected as a reference. All of the compressive strengths and flexural strengths of the specimens at 3, 7, and 28 days were lower than that of the OPC, especially the values of compressive strengths, these values were almost half of that value of the OPC. However, when the LRT addition was lower than 47% (C1), the values of the tested specimens met the MU15 standard in the autoclaved lime sand brick strength standard (GB/T 11945-2019); therefore, the specimen can be used for foundations and other buildings. It can also be used for the reclamation of the tailings pond, the hardening of the mining area, and the filling for mining wells in the mining area. Moreover, the values of the tested specimens were higher than that of the references [24,25]. After 28 days of curing, the compressive strength of the C1 specimen was about three times of that of the references. The additives were the municipal solid waste incinerators or the steel slag and the oil shale residue. The particle size distribution of the LRT and the references were almost the same, and also as other chemical compositions, except for the SiO_2_ content. The SiO_2_ content of the LRT was almost 5–10 times that of the references. The reason for the higher compressive strength could be attributed to the higher SiO_2_ content [26]. Furthermore, with the acid leaching, the calcium chloride was the main component of the LRT; the strength could be improved with the presence of calcium chloride during the curing process [27,28].

### 3.3. The Alkalinity of Cemented Building Materials

The pH values of hardened samples aged 28 days were measured, and the results were shown in Figure 2. The pH values were measured to be in the range of 9.99–12.52 and found to decrease with the increase in the LRT ratio. The slightly decreased pH value may be associated with the increased addition of LRT and other acidic components, which tends to reduce the strength properties of geopolymer binders [22]. In previous research, the polymerization process was best initiated at a high pH value, and the precipitation of portlandite may take place at a pH range of 9.5–12.97 [22]. Therefore, the LRT reaction with FA/OPC systems, which are synthesized under high alkaline concentration, results in a stable microstructure in the construction material.

### 3.4. Heavy Metal Leaching Risk of Cemented Building Materials

As shown in Table 2, the contents of the heavy metals were still high, and it was important to consider the possibility of solidifying heavy metals when the wastes were used as concrete materials [29]. The heavy metal contents that were leached out were low enough to meet the requirements of the Chinese GB5085.3-2007 [30] in terms of heavy metal content (Table 4). The addition of greater amounts of OPC, in turn, reduced the amounts of heavy metals that could be leached out. Heavy metals such as Cu, Zn, Cd, Pb, Cr, Hg, and Ni, are not only physical storage, but also special solidifying structures of adsorption and even bonding with matrix materials. Construction material solidification mechanisms were physical and chemical action; the physical effect was mainly physically attached, and the chemical effect formed chemical bonds in the cement reaction process [31]. This method of preparing building materials of LRT replaces OPC in the construction material preparation system and is of great significance to promote the sustainable development of the environment.

### 3.5. Discussion

The microstructures of LRT and specimens with 47% LRT added (C1) after 3, 7, and 28 days of curing are shown in Figure 3. The fracture of the polyester mortars is almost always caused by the failure of the fine aggregate [32], regardless of the type of filler. The particle size distribution had a significant relationship with the mechanical strength of the concrete [33], and the particle size of the leaching residue was relatively small when compared with tailings used in other research [34]. Moreover, the calcium sulfate contents of the leaching residue were obviously higher after the H_2_SO_4_ leaching process, an amorphous C–S–H gel coated the surface of the dehydrated grain, and the particles of the fillers and fine aggregates were sufficiently covered with a hardened polyester resin as a binder. It is evident that there was a high adhesion between the fillers or fine aggregate, and a polyester resin had developed. The hydrated product presented a denser structure and lower porosity after a longer curing time.

Thermal stability is an important indicator for the use of building materials. The TG analysis for the LRT and also the specimen (C1) are shown in Figure 4. When the temperature ascended to 1000 °C, the weight loss of LRT could reach 17.9%. There were exothermic peaks in the two temperature ranges of 760–820 °C and 930 °C–1000 °C, respectively. These two exothermic peaks correspond to a weight loss of 15.2%; the loss was caused by mineral decomposition and the gaseous substance produced.

The weight loss for the specimen (C1) was between 90–180 °C and 700 °C–1000 °C, respectively. The first loss was caused by the bound water volatilization; the loss was 1.9%. The second loss was 3.6%; it was also caused by the mineral decomposition and the unburned carbon decomposition of FA. However, the weight loss of the specimen was significantly smaller than that of the LRT. It shows that after curing and stabilization, the minerals in the specimen had reached a certain thermal stability equilibrium. This phenomenon also showed that the specimen has good thermal stability and can be applied to various environments [35]. When the stabilization index was tested, the carbonization coefficient and drying shrinkage were 0.79 and smaller than 0.42, respectively. It means the specimen has good stability.

The XRD comparison analysis was shown in Figure 5. It can be found that after curing, the main proportion of SiO_2_ in the LRT was converted into more stable minerals such as CaAl_2_ (SiO_3_)_4_ and Mullite. Thus, the good stability of the specimen could be obtained [18,36]. The solidified hydration reaction mixes the calcite with the partially crystalline gel, resulting in the coexistence of the amorphous phase and the crystalline phase of the hydrated product. This plays a crucial role in combining non-reactive and partially reactive particles to form a denser structure.

In order to assess the sustainability and environmental risks of the process, it is necessary to conduct an investigation based on the leaching toxicity of building materials. In this study, LRT was used to replace traditional building materials and stone materials. Therefore, compared with traditional cement, the use of this LRT mixing method can greatly reduce the impact on the environment so as to achieve no waste in the mining area.

## 4. Conclusions

Based on the characteristic of the leaching residue from Pb–Zn tailings, the curing method was chosen for its reuse of building materials. With the addition of LRT lower than 47%, the specimen had a better strength characteristic and fulfilled the MU15 class of the autoclaved lime sand brick strength standard; thus, the strength of the specimen can meet that of foundations and other buildings. The values of the tested specimens were about three times higher than that of the references. The high-strength phenomenon makes curing materials with LRT added have a wide range of applications, which not only effectively solves the tailing stacking problem but also solves the problem of the building material demands in the mining area. The TG analysis revealed that the specimen has good thermal stability; its carbonization coefficient and drying shrinkage were 0.79 and smaller than 0.42, respectively, and can be applied to various environments. Moreover, from the comparative analysis of the SEM and XRD, it can be inferred that the LRT and the additive formed stable minerals during the curing process so that the prepared test piece had good strength and good stability. Apart from the strength, the leaching concentrations of heavy metals such as Cu, Zn, Cd, Pb, Cr, Hg, and Ni decreased obviously after the curing process. The heavy metal contents that were leached out met the limits in GB5085.3-2007. The toxic leaching of heavy metals from the test specimens also met the relevant requirements, and its use in solidifying heavy metals of LRT is feasible. In addition, the cement materials of LRT can be applied to mine filling after more tests are conducted on the material fluidity, which will be further research. Based on the leaching recovery of the tailing, with wider reuse of the LRT, there will no more solid wastes generated.

## Figures and Tables

**Figure 1 materials-14-02940-f001:**
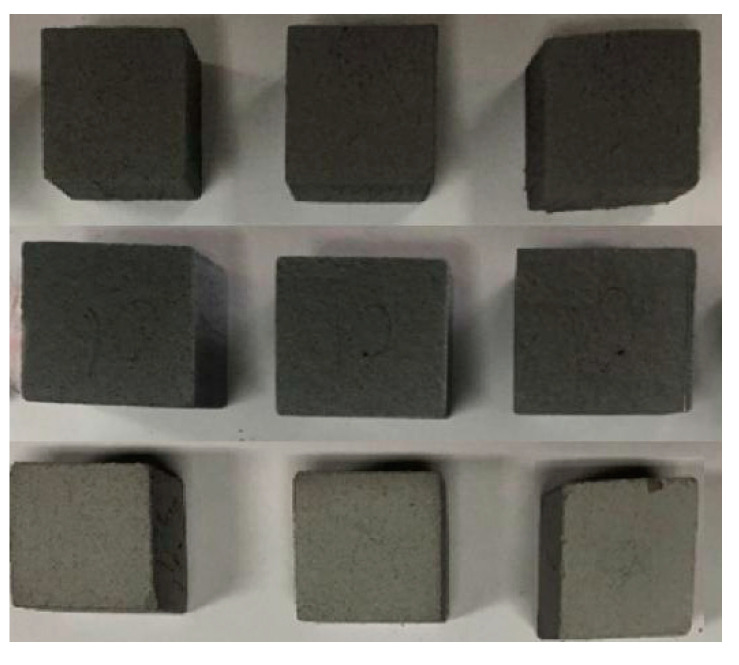
Pictures of solidified cemented building materials with different curing times.

**Figure 2 materials-14-02940-f002:**
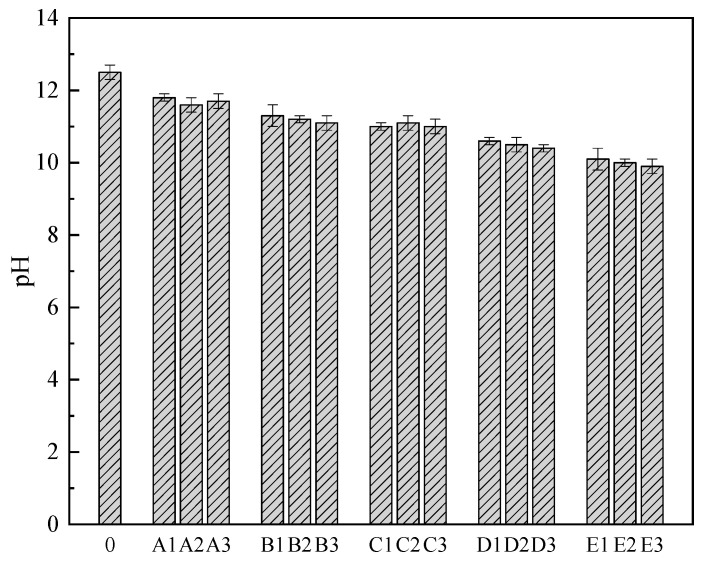
The pH values of designed harden samples with aged 28 days curing.

**Figure 3 materials-14-02940-f003:**
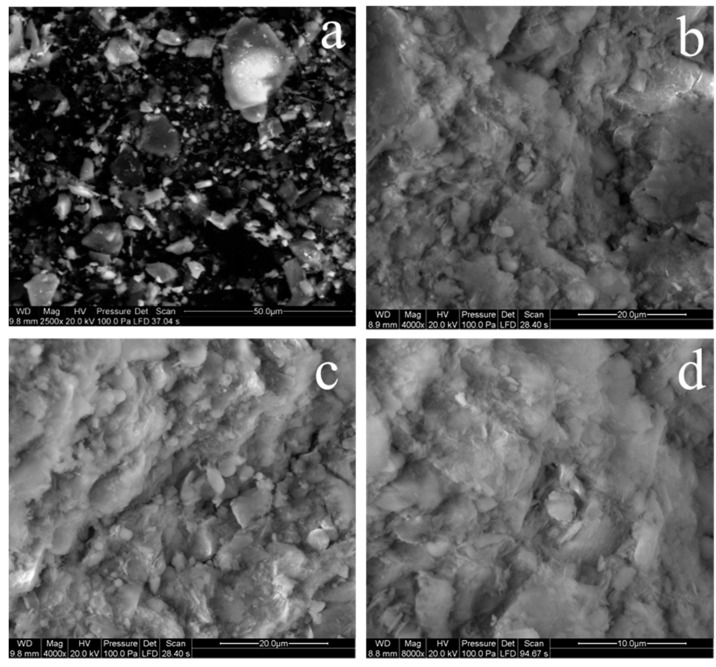
Microstructure of the LRT and the specimens with different hydrated curing times. (**a**), LRT; (**b**) C1 specimen with 3 days curing; (**c**), C1 specimen with 7 days curing; (**d**), C1 specimen with 28 days curing.

**Figure 4 materials-14-02940-f004:**
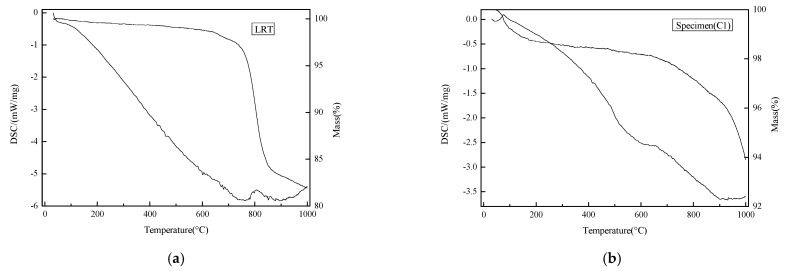
Thermogravimetric analysis of (**a**) LRT, (**b**) Specimen(C1).

**Figure 5 materials-14-02940-f005:**
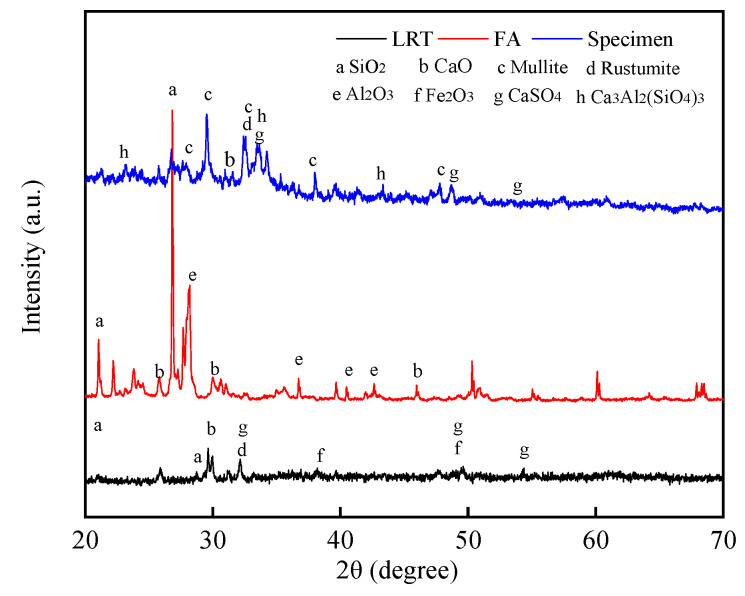
XRD Comparative Analysis.

**Table 1 materials-14-02940-t001:** Mix rate of LRT-FA-OPC.

Designation	Temperature (°C)	Humidity (%)	Rate (%)		
			OPC	FA	LRT
0	45	90	100	0	0
A1	45	90	70	3	27
A2	45	90	70	5	25
A3	45	90	70	7	23
B1	45	90	60	3	37
B2	45	90	60	5	35
B3	45	90	60	7	33
C1	45	90	50	3	47
C2	45	90	50	5	45
C3	45	90	50	7	43
D1	45	90	40	3	57
D2	45	90	40	5	55
D3	45	90	40	7	53
E1	45	90	35	3	62
E2	45	90	35	5	60
E3	45	90	35	7	58

**Table 2 materials-14-02940-t002:** Composition of the LRT.

Major Elements	Content (%)	Minor Elements	Contents (%)	Sulfur Forms	Proportion (%)
SiO_2_	73.86	Pb	0.0165	Sulfate	97.1
Fe_2_O_3_	3.66	Zn	0.06	Available sulfur	2.9
Al_2_O_3_	3.16	Cd	0.0001	-	-
CaO	2.72	Cr	0.0175	-	-
MgO	0.44	As	0.0050	-	-
Na_2_O	0.1	Hg	0.0001	-	-
K_2_O	0.16	Sn	0.0013	-	-
SO_2_	12.87	Ni	0.0002	-	-
P_2_O_5_	0.01	Cu	0.0046	-	-
TiO_2_O	0.78	-	-	-	-
LOI	17.21	-	-	-	-
TC	1.55	-	-	-	-

**Table 3 materials-14-02940-t003:** Specimen preparation and its mechanical strength properties after different curing days.

Designation	Compressive Strength (MPa)			Flexural Strength (Mpa)		
	3d	7d	28d	3d	7d	28d
0	21.61	37.71	40.69	4.67	5.35	5.90
A1	13.71	18.57	21.52	3.31	3.94	4.32
A2	14.08	19.05	24.31	3.62	4.21	4.49
A3	15.50	18.50	21.70	3.60	4.00	4.10
B1	10.90	17.70	20.90	3.60	4.12	4.20
B2	10.40	17.50	19.80	3.40	3.87	4.26
B3	9.10	16.45	18.12	3.17	3.55	4.02
C1	9.11	15.80	17.20	3.57	3.98	3.95
C2	9.35	16.20	16.59	3.22	3.55	3.35
C3	9.67	15.30	16.23	3.27	3.15	3.42
D1	6.71	10.56	10.30	0.54	1.26	1.72
D2	7.17	11.05	12.10	0.91	1.29	1.59
D3	7.04	11.03	11.20	0.88	0.82	1.45
E1	3.34	5.61	7.37	0.37	0.32	0.33
E2	4.28	4.03	7.23	0.46	0.48	0.64
E3	3.27	4.53	7.84	0.42	0.48	0.39
Standard MU15	-	-	15	-	-	3.3
[23]	-	-	5.71	-	-	-
[24]	-	-	4.24	-	-	-

**Table 4 materials-14-02940-t004:** Leaching of heavy metals from concrete materials.

HMs (mg/L)	Cu	Zn	Cd	Pb	Cr	Hg	Ni
LRT	2.35	23.1	0.04	3.74	0.001	0.002	0.07
FA	0.58	1.01	0.01	0.74	0.02	0.06	0.02
OPC	0.05	0.07	ND	0.02	ND	ND	0.01
C1	0.09	0.22	ND	0.12	ND	ND	ND
C2	0.08	0.24	ND	0.10	ND	ND	ND
C3	0.09	0.21	ND	0.11	ND	ND	ND
GB5085.3-2007	100	100	1	5	15	0.1	5

The detection limits for ICP-MS: Cu (0.009 ug/L), Zn (0.007 ug/L), Cd (0.006 ug/L), Pb (0.005 ug/L), Cr (0.005 ug/L), Hg (0.001 ug/L), Ni (0.001 ug/L). ND: under the detection limits.

## Data Availability

Research data can be obtained from the author.

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
