# Peer review of "Mechanical Properties and Toxicity Risks of Lead-Zinc Sulfide Tailing-Based Construction Materials"

_materials, 2021, doi:10.3390/ma14112940_

Round 1

Reviewer 1 Report

Manuscript ID: materials-1224635

Title: Compressive strength, Hardness and Toxicity risk of Construction Material synthetized from tailing Leaching Residue

Authors: Yang Zhou et al.

Title and Abstract must write about raw materials: Tailing Leaching Residue from what process?

I believe that authors must change Title to: Compressive strength, Hardness and Toxicity risk of Construction Material Synthetized from Leaching Residue of the Pb-Zn Sulfide Tailing

Line 33-34. Authors must write some sentences about leaching process. What optimum conditions and raw materials?

Authors must write more information about mineral composition of residue, its influence on physical properties of construction materials and write more exactly the novelty of the article and the difference between this study and previous researches.

Table 1. How were chose the LRT amount? Why uses these volume?

Table 2. What are the LOI (loss on ignition at 1000 ºC) and C (carbon) content? Use subscript numbers in the names of oxides (SiO2 etc.)

  1. Add chemical compositions of fly ash with LOI and carbon content.
  2. What the mineral compositions of LRT and FA (add XRD figure)?
  3. What are the average particle size distribution of this raw materials?
  4. Add figure for this data (don’t write information in article – section 3.2).

Table 3. Change this table to figure with columns. Add error bars.

Figure 3. Add the indicator of the figures - a, b, c, e. Figure 3a is taken at a different magnification. There are no differences between figures 3 b, c, e. Why?

Figure 4. Please provide a fuller description of figure 4 in the caption. Why are the TGAs given for only these two samples? Where is the TGA of fly ash? What is the peak in Figure 4a at 800 ºC? What does it correspond to phase transformation?

Figure 5. All peaks must be signed, why are angles 20-70 taken? Diffractograms with a lot of noise. Very bad quality. A longer exposure is needed.

Conclusions should be more specific and contain several points (3-5), the authors should add the results of the research (values and characteristics of the materials received)

Technical errors:

Change the reference style from (Chen et al., 2018) to [1] in all article.

Line 93-94. Change mineralogical properties to mineral compositions.

Line 97-98. Write full chemical formulas of nitric acid and sulfuric acid.

Line 99. Write full name of ICP-MS.

References are not in Materials style.

In this from article is not corresponding to Materials level and should be rejected.

Author Response

Thank you, the MS was rechecked and revised according to your valuable suggeation as the attachment.

Reviewer 2 Report

Dear authors,

This research assessed the use of secondary resources in construction materials production. This paper has relevant data, but it should be further improved. Here are some questions and suggestions.

  1. The title has a lot of information. I suggest turning the title in a more concise one.
  2. In the abstract, line 21 and 22, How “excellent” construction materials and their properties in terms of strength and stability can be? Introduce quantitative information to specify the benefits of using LRT in construction materials.
  3. Abbreviations should be avoided in the abstract section.
  4. The abstract should include the main outcomes of the research conducted in a quantitative manner.
  5. Keywords should be more targeted (e.g. mine tailings, leaching, mining, construction material, compressive strength, chemical stability)
  6. In the introduction, line 33, how huge is the quantity of tailings production? Present an estimation.
  7. There exist multiple references which should be avoided, for example: (Shu et al., 2018; Chen et al., 2018; 47 Terzic et al., 2012; Liu et al., 2019). Please check the manuscript and eliminate ALL the lumps in the manuscript. Each reference must be characterised individually, by mentioning 1 or 2 phrases per reference to show how it is different from the others and why it deserves mentioning. Multiple references are of no use for a reader and can substitute even a kind of plagiarism, as sometimes authors are using them without proper studies of all references used. 
  8. Generally, the introduction is incomplete. More data about mining is missing, as well as the potential to use LRT in construction materials, their physical-chemical properties, and comparisons with other works. The problem addressed in the research should also be deeper explored and the increase need to reuse and recovery secondary resources in closed loop systems should be emphasize. In addition, since the authors mentioned a sustainable approach, the specific Sustainable Development Goals from United Nations, that are being tackled with this research, could be mentioned. Bearing in mind the climate emergency, clarify the reader why high carbon emissions are associated to OPC production and why is important to find substitutes.
  9. In the Experimental section, the authors should mention in 2.1 where the sample was collected. The city and country should be included, and, if possible, the geographic coordinates.
  10. In section 2.2, the authors should mention which standard or protocol was followed to produce the construction materials.
  11. The authors present results and discussion in section 3. Section 3 should be renamed as “Results and Discussion” and section 4 should be merged with section 3.
  12. In line 113, 118 and 142, correct the chemical forms “SiO2”, “Al2O3” and “SO2” (subscript the numbers).
  13. In figure 1, include the curing time in the caption.
  14. Concerning table 3 data, what are current feasible applications of the construction materials produced? Include an overview on this.
  15. In line 166, Material should be in lowercase.
  16. In table 4, include a footnote with the detection limits of the elements analysed. Also, specify what is ND in the footnote.
  17. Generally, Results and Discussion section is too descriptive. Comparisons with other works are missing, as well as other explanations that support the behaviors observed. This section needs to be improved.
  18. Conclusions could be improved – How did your research help to tackle the problem described? What further developments should be considered? Include an overview on these topics.
  19. English writing must be improved.

Author Response

Thank you very much, the MS was revised according to your valuable suggestions.

Round 2

Reviewer 1 Report

The authors have significantly improved the quality of the article. The authors made changes in all parts including title, abstract, introduction, materials and methods results, conclusions and references.

This article "Compressive strength, Hardness and Toxicity risk of Construction Material Synthetized from Leaching Residue of the Lead Zinc Sulfide Tailing" can be accepted in Materials.

Author Response

Thanks for your valuable suggestions!

Reviewer 2 Report

Dear authors,

The manuscript needs to be improved to be in an acceptable form for publication:

  1. The title has a lot of information and it is too long. I suggest turning the title in a more concise one (e.g Mechanical properties and toxicity risks of Lead-Zinc Sulfide Tailing-based construction materials)
  2. The introduction is incomplete. More data about mining is missing, as well as the potential to use LRT in construction materials, their physical-chemical properties, and comparisons with other works. The problem addressed in the research should also be deeper explored and the increase need to reuse and recovery secondary resources in closed loop systems should be emphasize. In addition, since the authors mentioned a sustainable approach, the specific Sustainable Development Goals from United Nations, that are being tackled with this research, must be mentioned. Bearing in mind the climate emergency, clarify the reader why high carbon emissions are associated to OPC production and why is important to find substitutes.
  3. Results and Discussion section is too descriptive. Comparisons with other works are missing, as well as other explanations that support the behaviors observed. This section needs to be improved.
  4. In conclusions, the authors should explain how their research helped to tackle the environmental problems, that are related to mine tailings and construction products, and concerning the climate emergency. Also, further developments should be specified.

Author Response

Thanks for your valuable suggestions!
